# Perry Disease: Current Outlook and Advances in Drug Discovery Approach to Symptomatic Treatment

**DOI:** 10.3390/ijms251910652

**Published:** 2024-10-03

**Authors:** Zbigniew Gajda, Magdalena Hawrylak, Jadwiga Handzlik, Kamil J. Kuder

**Affiliations:** Department of Technology and Biotechnology of Drugs, Faculty of Pharmacy, Jagiellonian University Medical College in Kraków, Medyczna 9, 30-688 Krakow, Poland; zbigniew.gajda@student.uj.edu.pl (Z.G.); magdalena.hawrylak@student.uj.edu.pl (M.H.); j.handzlik@uj.edu.pl (J.H.)

**Keywords:** Perry disease, neurodegeneration, rare disease, pharmacophore, MAO-B, SERT, serotonin 5-HT_1A_ receptor, dopamine D_2_ receptor, polypharmacology, multi target drug

## Abstract

Perry disease (PeD) is a rare, neurodegenerative, genetic disorder inherited in an autosomal dominant manner. The disease manifests as parkinsonism, with psychiatric symptoms on top, such as depression or sleep disorders, accompanied by unexpected weight loss, central hypoventilation, and aggregation of DNA-binding protein (TDP-43) in the brain. Due to the genetic cause, no causal treatment for PeD is currently available. The only way to improve the quality of life of patients is through symptomatic therapy. This work aims to review the latest data on potential PeD treatment, specifically from the medicinal chemistry and computer-aided drug design (CADD) points of view. We select proteins that might represent therapeutic targets for symptomatic treatment of the disease: monoamine oxidase B (MAO-B), serotonin transporter (SERT), dopamine D_2_ (D_2_R), and serotonin 5-HT_1A_ (5-HT_1A_R) receptors. We report on compounds that may be potential hits to develop symptomatic therapies for PeD and related neurodegenerative diseases and relieve its symptoms. We use Phase pharmacophore modeling software (version 2023.08) implemented in Schrödinger Maestro as a ligand selection tool. For each of the chosen targets, based on the resolved protein–ligand structures deposited in the Protein Data Bank (PDB) database, pharmacophore models are proposed. We review novel, active compounds that might serve as either hits for further optimization or candidates for further phases of studies, leading to potential use in the treatment of PeD.

## 1. Introduction

### 1.1. Neurodegeneration and Inflammation

Neurodegenerative diseases (NDDs) constitute a diverse set of neurological disorders that impact millions of people globally, leading to the gradual deterioration of the nervous system. The vast spectrum of NDDs includes Alzheimer’s disease (AD), Parkinson’s disease (PD), primary tauopathies, frontotemporal dementia (FTD), amyotrophic lateral sclerosis (ALS), synucleinopathies (i.e., Lewy body dementia [LBD] and multisystem atrophy [MSA]), Huntington’s disease (HD) and related polyglutamine (polyQ) diseases (including spinocerebellar ataxias [SCA]), prion disease (PrD), traumatic brain injury (TBI), chronic traumatic encephalopathy (CTE), stroke, spinal cord injury (SCI), and multiple sclerosis (MS) [1].

Worldwide, NDDs impact millions of individuals. The two most prevalent NDDs are PD and AD. A 2024 report from the Alzheimer’s Disease Association estimated that as many as 6.9 million patients in the USA may suffer from AD [2]. On the other hand, the Parkinson’s Foundation estimates that approximately one million patients in the USA are diagnosed with PD [3].

NDDs share many fundamental processes associated with progressive neuronal dysfunction and death, including oxidative stress, programmed cell death, proteotoxic stress, and its attendant abnormalities in the ubiquitin–proteasomal and autophagosomal/lysosomal systems, and neuroinflammation [4]. These processes cause the deterioration of neural networks in either the central (CNS) or peripheral (PNS) nervous system, which ultimately results in impaired memory, cognition, behavior, sensory perception, and/or motor function [1]. Consequently, the clinical presentations of NDDs can be used to categorize them broadly, with the most common types being extrapyramidal and pyramidal movement disorders, as well as cognitive or behavioral disorders. Most patients have a combination of clinical features, with very few having pure syndromes. Hence, neuropathological evaluation during the autopsy constitutes the gold standard for diagnosis, as specific protein accumulations and anatomic vulnerability are typically used to define NDDs [4].

While several medications are currently approved to treat NDDs, most of them provide only symptomatic treatment. The blood–brain barrier’s (BBB’s) limiting properties, which prevent nearly 99% of all xenobiotics from entering the brain, are the main cause of the lack of pathogenesis-targeting treatments [5].

### 1.2. Rare Diseases

A rare or orphan disease is a medical condition that affects a small percentage of the population [6]. As of 2021, rare diseases affect more than 470 million people worldwide—approximately 1/16 of the global population [7]. Despite significant advances in research, which have enhanced our understanding of the molecular foundations of these diseases and the availability of regulatory and economic incentives to speed up the development of treatments, most rare diseases still lack approved therapies [8]. One of the primary challenges in treating these conditions is the lack of standardized terminology and definitions, which hampers accurate diagnosis, disease classification, and the development of targeted treatments. Regulatory agencies provide incentives for pharmaceutical companies to develop therapies for these conditions, known as orphan drugs [9]—e.g., the Food and Drug Administration (FDA) grants orphan drugs sponsors tax credits for qualified clinical trials, exemption from user fees, or a potential seven years of market exclusivity after approval [10,11]. Still, less than 10% of patients with rare diseases receive treatments specifically tailored to their conditions [7]. Developing orphan drugs involves various strategies, including protein replacement therapies, small-molecule therapies, gene and cell therapies, and drug repurposing. Each approach comes with its strengths and limitations, and the process is further complicated by challenges in clinical trials, such as difficulties in patient recruitment, incomplete understanding of disease mechanisms, increased genetic heterogeneity, lack of animal models, and ethical concerns, particularly in pediatric cases. Additionally, the legislative procedure does not differ significantly from registering medicines for more common diseases—it adds another level of complexity to developing treatment for rare diseases, but on the other hand, grants necessary safety to patients upon releasing the drug to the market. Overcoming these barriers requires a collaborative effort involving academic institutions, industry, patient advocacy groups, and regulatory bodies to ensure that advances in rare disease research can be effectively translated into viable treatments [8].

### 1.3. Perry Disease

Perry disease (PeD) is a rare, genetic NDD inherited in an autosomal dominant manner. The disease manifests as parkinsonism, with psychiatric symptoms on top, such as depression or sleep disorders, and is accompanied by unexpected weight loss, central hypoventilation, and aggregation of DNA-binding protein (TDP-43) in the brain [12,13].

The cause of PeD is a mutation in the dynactin I gene (DCTN1), which is responsible for encoding the p150 subunit. Dynactin is a motor protein associated with axonal transport, while the aforementioned subunit constitutes a microtubule-binding site, an important feature of dynactin action [12]. Up until the fall of 2023, over 30 families with PeD have been reported [14]. Other than the “classic” type of disease, distinct phenotypes are recognized and classified as PeD [12].

### 1.4. Perry Disease Treatment

Due to the genetic cause, no causal treatment for PeD is currently available. Therefore, for patients suffering from this condition, the only way to improve their quality of life is through symptomatic therapy [15].

In this context, lines of evidence indicate levodopa (L-DOPA) [16,17,18], monoamine oxidase B (MAO-B) inhibitors [17], dopamine agonists [19,20], L-DOPA decarboxylase inhibitors [21], anticholinergics [17] or, very generally, wide-ranging groups of antidepressants, e.g., selective serotonin reuptake inhibitors (SSRIs) and tricyclic antidepressants (TCAs) [17,18], as helpful for PeD patients. Yet, to the best of our knowledge, no specific treatment for PeD has been either proposed or approved by any of the relevant legislative bodies.

## 2. Aim of Work

This work aims to review the latest data on potential PeD treatment, specifically from the medicinal chemistry and computer-aided drug design (CADD) points of view. Based on the medication therapies described so far and our knowledge, we have selected proteins that might represent therapeutic targets for the symptomatic treatment of the disease. The targets of focus in this work break down as follows: enzymes—MAO-B, sodium-dependent serotonin transporter (SERT); receptors—dopamine D_2_ (D_2_R) and serotonin 5-HT_1A_ (5-HT_1A_R). We will report on compounds that may be potential hits for developing symptomatic therapies for PeD and related NDDs, for relieving symptoms.

Since the selected proteins have been widely explored as potential therapeutic targets and, consequently, the vast chemical space of the ligands has been proposed for them, in the present work, we have mostly focused on the structures published since 2020, which, in preliminary biological studies, display expected activity toward the objectives of the review. 

To narrow down the search area, we use the pharmacophore modeling software Phase implemented in Schrödinger Maestro [22,23]. For each of the chosen targets, based on the resolved protein–ligand structures deposited in the Protein Data Bank (PDB) database (Table 1) [24], pharmacophore models are proposed using the default settings, except for SERT and 5-HT_1A_R, for which the following features have been manually added: positive ionic, aromatic ring, and hydrophobic or another aromatic ring (respectively). In the next step, the appropriately filtered (parameters given at the beginning of each section) ligand databases downloaded from ChEMBL [25,26] are screened using the proposed model. Up to the 10 most favorable results, in our opinion, are then tabulated (ranked by the Phase Screen Score value) and shortly described. The majority of the compounds from the screening results exhibit all the pharmacophore model features (for their respective target), except the D_2_R ligands, which mostly lack one of the aromatic features (the raw screening results are available as Appendix A).

## 3. Results

### 3.1. Enzymes

#### 3.1.1. Monoamine Oxidase B

Monoamine oxidase (MAO) is a mitochondrial enzyme that catalyzes the oxidative deamination of various monoamines. It plays a significant role in the metabolism of released neurotransmitters and the detoxification of a wide variety of endo- and exogenous amines. Two isoforms of this enzyme that are approximately 70% identical to each other are known—monoamine oxidase A (MAO-A) and MAO-B. MAO-A is the predominant form in the gastrointestinal tract, placenta, and heart, while MAO-B is prevalent in brain glial cells and platelets. Regardless of the isoform or occurrence, both are covalently bound to flavin adenine dinucleotide (FAD) [31,32].

Many studies suggest that MAO-B participates in the pathomechanism of NDDs associated with aging. Unlike most enzymes, its activity does not decrease but increases linearly beyond 60 years of age. The enzyme is also considered to be involved in the formation of free radicals. Due to its function, MAO-B is also the main enzyme involved in dopamine metabolism, therefore playing a key role in the pathophysiology of PD. Hence, MAO-B inhibitors in combination with levodopa have found use in PD treatment [33].

MAO-B inhibitors have excellent efficacy and are safe for use both in the initial stages of PD and (as adjunctive therapy) in its advanced form. Longer exposure to MAO-B inhibitors results in a lower demand for levodopa and slower disease progression. Drugs currently approved for therapy include irreversible MAO-B inhibitors selegiline and rasagiline and the reversible inhibitor safinamide (Table 2) [34].

Out of the structure of MAO-B complexed with safinamide (inhibitor; PDB ID: 2V5Z), a pharmacophore model was proposed (Figure 1 and Figure 2, Table 3).

#### 3.1.2. 2V5Z (Monoamine Oxidase B) Pharmacophore Screening

In the ChEMBL database, we queried for molecules exhibiting a half maximal inhibitory concentration (IC_50_) ≤ 100 nM in human MAO-B inhibition in assays published in 2020 and later, then conducted pharmacophore-based ligand screening using the model proposed by the Phase module (Table 4).

Seven out of ten compounds screened were published within the same work [38] focusing on fragment-based drug design (FBDD) for the discovery of selective MAO-B inhibitors. In this study, a steric clash-induced binding allosteric (SCIBA) strategy was used, in which the fragment entering the collision with the non-biological target protein—in this case, MAO-A—was the pharmacophore element. This arrangement provided greater selectivity to the correct target, MAO-B. Based on the structure of safinamide, the researchers found the fragment that was most sterically unfavorable for MAO-A (1-fluoro-3-phenoxymethylbenzene) and looked for combinations with fragments that could match the MAO-B active site, e.g., safinamide forms hydrophobic interactions with Phe103, Leu164, Leu167, Leu171, Ile199 and Tyr398, and it forms a hydrogen bond with Gln206. In the case of MAO-A, the binding site is curved so that safinamide collides sterically with Phe208. This results in an unfavorable conformational change of safinamide and a lack of hydrogen bonding with Gln215, which significantly worsens the affinity for this enzyme [38]. Based on these observations, a set of (S)-2-(benzylamino)-propanamide derivatives were designed, synthesized, and biologically evaluated. Two series of compounds were obtained. The first series included compounds **M4**, **M7,** and **M9**—and it was devoid of heterocyclic moieties (except for CHEMBL4759613 containing morpholine, not listed herein). The second series containing azacyclic amides included **M1**, **M2**, **M3**, and **M5**. Modifications of safinamide involving the addition of fluoride or a methyl group into central benzyl position 2 achieved strong inhibitory activity against MAO-B. The best activity among those reported was achieved by the **M4** containing chloride substituent in position 2 of the central benzene ring, which ranked fifth in our screening [38].

Studies of a series of azacyclic amides also showed that the presence of a chiral group is beneficial for MAO-B inhibition, i.e., the MAO-B inhibitory activity of S-enantiomer **M3** (IC_50_ = 21 nM) was superior when compared to its racemate **M2** (IC_50_ = 46 nM) and similar to that of safinamide. **M1** and **M5** with electron-acceptor substituents (-F or -Cl) also showed strong inhibitory activity (IC_50_ = 26 nM and 35 nM, respectively) [38]. The strongest activity of all seven compounds was shown by **M3**, which was ranked third in our screening. **M1**, with slightly less MAO-B inhibitory activity, was identified as the best fit by our model [38].

**M6** and **M8** were obtained using the FBDD method, based on a previously described series of (S)-2-(benzylamino)propanamide derivatives, which led them to conclude that the chiral amide group located at position 2 of the azetidine ring was important for MAO-B inhibition [39]. The modifications involved the introduction of chiral fluorinated pyrrolidine derivatives into a new series of compounds. **M8** appeared to be the most active, having a chiral fluorine atom in position 4 of the pyrrolidine ring. **M6**, obtained by introducing a fluorine atom into position 2 of the benzene ring, also showed good MAO-B inhibitory activity. Both compounds also showed remarkably high selectivity for MAO-B over MAO-A (**M6**, MAO-A IC_50_ = 29360.0 nM, **M8**, IC_50_ = 46365.0 nM) [39].

Last in order, according to our screening, **M10** was derived from the study, which was a continuation of the search for MAO-B inhibitors using FBDD methods [40]. Previously, the researchers combined rasagiline with a hydrophobic molecule, resulting in selective compounds with promising activities against MAO-B, and in the study described here, the linkers and hydrophobic groups were modified to yield compounds with a 1-(prop-2-yn-1-ylamino)-2,3-dihydro-1H-inden-4-thiol scaffold [40].

**M10** with a 3-(trifluoromethyl)benzyl substituent had very high activity (IC_50_ = 4.7 nM) and was highly selective for MAO-B over MAO-A (Selectivity Index [SI] = MAO-A IC_50_/MAO-B IC_50_ = 1641.3). These were better results than those achieved by rasagiline and safinamide in the same study. The only compound with even higher activity against MAO-B and selectivity, as obtained in this study, had a 1-methyl-3-propylbenzene fragment instead of a 1-ethyl-3-(trifluoromethyl)benzene fragment (IC_50_ = 0.35 nM, SI = 14162.9) [40].

In light of this review, it can be deduced that compounds containing, from the left, a 1-fluoro-3-((p-tolyloxy)methyl)benzene fragment linked to an amine or azacyclic ring and an amide group (located on the right side of the compound) with halogen substituents or a methyl group at position 2 of the central aromatic ring have the potential to be potent MAO-B inhibitors. Further, the chirality of the halogen group on the azacyclic ring is of relevance to SAR for the series of described compounds. Last but not least, the (2,3-dihydro-1H-inden-4-yl)(3-(trifluoromethyl)benzyl)sulfane fragment linked to the amine-alkyne fragment is a promising framework for study and further modification.

#### 3.1.3. Sodium-Dependent Serotonin Transporter

SERT is a protein located in presynaptic neurons. The raphe nuclei’s presynaptic neurons release serotonin, which activates the limbic system. Then, serotonin attaches itself to postsynaptic serotonin receptors, which are mostly found in limbic regions like the nucleus accumbens, dorsal striatum, hippocampal regions, and cortex. One of the processes for taking the neurotransmitter out of the synaptic cleft is serotonin reuptake, which is facilitated by SERT (Figure 3) [41].

Serotonin is related to the regulation of social behavior and emotional responses. Disturbances in serotonin transmission are related to depressive symptoms such as feelings of profound sadness, worthlessness, low self-esteem, suicidal thoughts, and lowered cognitive abilities [41]. Multiple medications achieve an increase in the serotonin concentration in the synaptic cleft by stabilizing the inactive state of SERT, thereby having an antidepressant effect [43]. While serotonin does not play much of a role in PD motor symptoms, serotonergic dysfunction is relevant to PD nonmotor symptoms, like depression, fatigue, weight changes, and visual hallucinations. While the first two are related to inhibition, the latter, on the contrary, is related to an increase in serotonergic transmission [44]. Antidepressants can alleviate them all, but the data on reducing psychotic symptoms are of poor quality [45]. Numerous antidepressants possess SERT activity as well (Table 5) [46].

Out of the structure of SERT complexed with sertraline (inhibitor; PDB ID: 6AWO), a pharmacophore model was proposed (Figure 4 and Figure 5, Table 6).

#### 3.1.4. 6AWO (Sodium-Dependent Serotonin Transporter) Pharmacophore Screening

In the ChEMBL database, we queried for molecules exhibiting an IC_50_ ≤ 100 nM in human SERT inhibition in assays published in 2020 and later, then conducted pharmacophore-based ligand screening using the model proposed by the Phase module (Table 7).

Except for established, well-known medicines, the screening output described herein contained compounds from three separate studies.

As a novel compound with the highest PhaseScreenScore, there appeared **S1** [49], developed in a study exploring a novel dual receptor for advanced glycation end products (RAGE)/SERT inhibitors for potential application in treating co-morbid AD and depression. Combining such dual inhibition could be beneficial in both conditions, since RAGE facilitates β-amyloid neuronal damage, and its blockade can notably prevent β-amyloid-induced neurotoxicity. The authors based their novel molecules on fusing vilazodone and azeliragon structures: SERT and RAGE inhibitors, respectively (Table 8). Analysis of the potential binding modes of azeliragon to RAGE and vilazodone to SERT showed that, between the aminoalkyl azeliragon moiety and benzofuran vilazodone moiety and their targets, there exists an adjacent pocket. Additionally, the imidazole azeliragon moiety and benzofuran vilazodone moiety (both aromatic heterocycles) could form bonds and interactions with their respective targets, which signifies that central aromatic heterocycles are common, crucial pharmacophoric features in these compounds. Based on these findings, the key pharmacophore structures of both compounds were fused [49].

Firstly, the synthesized chimeric **S10** exhibited some inhibitory potential on both RAGE and SERT, without inheriting vilazodone’s partial agonism toward 5-HT_1A_R. Sadly, it exhibited serious cytotoxicity (which was the reason azeliragon was withdrawn from phase III clinical trials). Because of that, structural modifications were proposed to improve its bioactivity and safety.

Out of pyrazole, phenylimidazole, and thiazole, the only central heterocyclic moiety that preserved the RAGE and SERT activities was the thiazole; therefore, **S10** and **S11** were subjected to further modifications. It is worth noting that the thiazole derivatives preserved dual inhibition better than the pyrazole or benzimidazole derivatives, and the imidazole derivatives displayed stronger SERT inhibition than the thiazole derivatives.

Exploring different substituents on the thiazole core structure showed that, out of the alkyl substituents on thiazole, n-butyl was superior to methyl, ethyl, propyl, and isopropyl for RAGE inhibition, and out of the n-alkyl linkers between piperidine and indole, n = 4 was superior for dual inhibition.

Exploring different substituents on the imidazole core structure showed that, of the alkyl substituents on imidazole, ethyl, propyl, n-butyl, and cyclobutyl were superior to n-pentyl, isopropyl, cyclopropyl, cyclopentyl, cyclohexyl for RAGE inhibition, and out of the n-alkyl linkers between piperidine and indole, n = 1 and n = 4 were superior to n = 2 and n = 3 for dual inhibition.

Out of the n-substituents on indole, H or methyl was most suitable, as increasing the size of the substituent decreased the dual inhibition.

Molecular-docking simulations to RAGE and SERT showed that **S12’s** calculated pose was almost consistent with the calculated poses of the reference compounds (azeliragon and vilazodone) in their respective targets, with somewhat retained interactions, thus endorsing its biological activity. Furthermore, it had a better neuroprotective effect against β-amyloid_25–35_ than azeliragon and substantially lowered the immobility time in the tail suspension test, which indicates a potential antidepressant effect, yet was less potent than vilazodone. Although **S12** is the most promising molecule highlighted by the authors, it has not been returned by 6AWO-based pharmacophore screening. 

In summary, **S12**, which is a first-generation dual RAGE/SERT inhibitor, has demonstrated the viability of the pharmacophore fusion strategy and offered a useful prototype for the possible treatment of AD with comorbid depression [49].

The next four novel compounds with the highest PhaseScreenScore, which emerged from the 6AWO-based pharmacophore screening, were **S2**–**S5**, along with dextromethorphan, which was the baseline structure of the novel compounds (Table 9) [48].

Dextromethorphan is a commonly used medicine, mainly as a cough suppressant, co-administrated with quinidine for the treatment of pseudobulbar affect and recently co-administered with bupropion for the treatment of major depressive disorder [53].

Dextromethorphan, like other aryl-methyl ethers, is subjected to in vivo O-dealkylation, yielding dextrorphan, which through N-methyl-d-aspartic acid receptor (NMDAR) inhibition may cause dissociative hallucinations when consumed in an excessive amount. As it facilitates dextromethorphan recreational use, efforts have been made to formulate a dextromethorphan analogue that is unusable for recreational use while still retaining the desirable pharmacological action [48]. Since dextromethorphan (co-administrated with bupropion) is indicated in the treatment of major depressive disorder and was also found (co-administrated with quinidine) to benefit levodopa-induced dyskinesia in PD, thus exhibiting valuable performance in treating other neurological and psychiatric diseases, it might be beneficial to explore its analogues [54,55]. This is further supported by the fact that AVP-786, its deuterated analogue, was studied in clinical trials (co-administrated with quinidine) for CNS disorders as well.

Common strategies to prevent O-dealkylation include fluorination and deuteration. Dextromethorphan may be fluorinated by replacing the aryl methyl ether with various fluoroalkyl ethers or fluoroalkyls. Overall, fluorinated dextromethorphan analogues sustained dextromethorphan’s pharmacological profile, while having slightly weaker affinity to the sigma_1_ receptor (σ_1_R), sustaining affinity to the sigma_2_ (σ_2_R) receptor and ceasing affinity to NMDAR. Surprisingly, **S3** gained an affinity for sodium-dependent noradrenaline transporter (NET) (IC_50_ = 944 nM). Additionally, **S2**–**S5** also gained an affinity for SERT. Fluorinated analogues also maintained similar pharmacochemical properties compared to dextromethorphan, namely high aqueous solubility, while simultaneously improving the in vivo pharmacokinetics [48]. In comparison, deuteration did not show an influence on the pharmacokinetics and other drug-like properties compared to dextromethorphan. The selectivity and affinity to receptors exhibiting neuropsychiatric effects seem to also be unchanged, aside from blocking metabolism to dextrorphan, which blocks the ability to antagonize NMDAR [56]. Overall, fluoroalkylated and deuterated dextromethorphan analogues seem to be promising future therapeutic options for the treatment of CNS disorders, especially Parkinson-like disorders [48,57].

The last four compounds in the screening results were described in the 2022 patent for ibogaine and its analogues as therapeutics for neurological and psychiatric disorders, and the compositions and methods for treating psychiatric disorders or their symptoms were considered (Table 10) [51]. 

Ibogaine is an indole alkaloid, naturally occurring in *Tabernanthe iboga*, a shrub native to Central–West Africa. It is an unusual psychedelic substance that can cause vivid memory recall and replay as well as oneirogenic effects, which are states akin to waking dreams. While high doses of ibogaine are used for their hallucinogenic effects during religious rituals and initiation rites, low doses are used as stimulants to prevent fatigue on hunting excursions and to dull hunger and thirst. Ibogaine is effective in interrupting drug dependence by providing quick and long-lasting relief from cravings and withdrawal symptoms in anecdotal reports and open-label case studies involving individuals addicted to heroin and cocaine. First-pass metabolism quickly demethylates ibogaine into the long-acting metabolite noribogaine [51,58]. Ibogaine and noribogaine bind with modest affinity to a variety of targets, including transporters, SERT, NET, sodium-dependent dopamine transporter (DAT), and receptors, opioid, acetylcholine (Ach), σ and NMDA [59]. Based on the ibogaine analogues, which were the results of the screening, switching methoxy slightly increases the inhibition of vesicular monoamine transporter 2 (VMAT2), and greatly SERT. Phenyl analogues exhibit greater inhibition of VMAT2 and SERT. N-methylation of pyrrole also potentiates the inhibition of both VMAT2 and SERT.

Summarizing the above-mentioned studies, it can be concluded that compounds containing a 3-(4-(4-(1-(4-(4-chlorophenoxy)phenyl)-1H-imidazol-4-yl)piperidin-1-yl)butyl)-1H-indole-5-carbonitrile backbone or one in which the imidazole site is occupied by a thiazole showed good SERT inhibitory activity (and some RAGE inhibitory activity on top). For SERT inhibition, the presence of an alkyl-substituted imidazole is most favorable (in particular, the ethyl and cyclobutyl substituents). Dextromethorphan derivatives with alkyl or alloxyfluoro substituents have also achieved good activities against SERT. The noribogaine backbone gains SERT inhibitory activity upon N-methylation of the pyrrole and the conversion of the -OH group to a cyanide substituent.

### 3.2. Receptors

#### 3.2.1. Dopamine Receptors

Dopamine is a catecholamine neurotransmitter that fulfills essential functions in both the CNS and PNS and is responsible for numerous effects: the inhibition of prolactin production, movement, behavior, motivation, the reward system, cognitive abilities including learning, attention, working memory, mood and even sleep. Dopamine acts via five dopamine receptors (D_1_, D_2_, D_3_, D_4,_ and D_5_) belonging to the G-protein-coupled receptors (GPCRs). Among them, two subclasses can be identified: the dopamine D_1_-like family, which includes D_1_ and D_5_ receptors, and the D_2_-like family, with D_2_, D_3_, and D_4_ receptors. Types 2, 3, and 4 share only a similar chemical structure, while types 1 and 5 also have similar drug sensitivity. The D_1_-like group are mostly postsynaptic receptors, binding mainly to the stimulatory G_s_ protein, while the D_2_-like receptors are involved as both postsynaptic receptors and presynaptic autoreceptors that bind to the inhibitory G_i/o_ protein [60,61].

##### Dopamine D_1_-like Family Receptors

D_1_-like receptors are found primarily in the cerebral cortex, the striatum, and the limbic system of the brain. In addition, they are also present in the cardiovascular system, as well as taking part in the regulation of neuronal growth. D_1_-like receptors are the most widespread of all the dopamine receptors in the human nervous system. D_1_-like receptors also show an impact on behavior, with roles including impulse control and involuntary movements, sleep, effects on learning and working memory, the reward system, and even the growth regulation and renin control in the kidneys [60].

When dopamine binds to D_1_-like receptors, guanosine nucleoside-binding proteins are activated, adenylyl cyclase activity is stimulated and, as a result, a cyclic AMP (cAMP) molecule is generated, acting as a secondary messenger. Other signaling pathways affect phospholipase C and calcium ion release. In the kidney and striatum, D_1_-like receptors through the protein kinase A and C signaling pathways also affect adenosine 5′-triphosphatase (ATPase) inhibition [60].

##### Dopamine D_2_-like Family Receptors

D_2_-like receptors are expressed in high concentrations in the olfactory bulb, substantia nigra, ventral tegmental area (VTA), putamen, caudate and nucleus accumbens. In small concentrations, they can be also found in the circulatory system, kidneys, gastrointestinal tract, cerebral cortex, hypothalamus, sympathetic ganglia, septum, and adrenal glands. Unlike D_1_-like receptors, D_2_-like receptors inhibit the activity of adenylate cyclase and cause a decrease in the cAMP concentration [60]. 

Most dopamine receptor agonists approved for therapy are D_2_-like receptors agonists (Table 11) [62,63]. These can be divided into ergoline, bromocriptine, and pergolide (withdrawn from human use by the FDA [64], it has some affinity for D_1_R,), and non-ergoline derivatives: pramipexole, ropinirole, and rotigotine. Apomorphine is a less specific agonist and acts on all the dopamine receptors, although mainly on D_2_-like receptors [65].

Out of the structure of D_2_R complexed with rotigotine (agonist; PDB ID: 8IRS), a pharmacophore model was proposed (Figure 6 and Figure 7, Table 12).

#### 3.2.2. 8IRS (Dopamine D_2_ Receptor) Pharmacophore Screening

In the ChEMBL database, we queried for molecules exhibiting a half maximal effective concentration (EC_50_) ≤ 100 nM of human D_2_-like receptor activation in assays published in 2020 and later, then conducted pharmacophore-based ligand screening using the model proposed by the Phase module (Table 13).

Three of the ten compounds resulting from our screening (**P1**, **P4**, and **P7**) appeared in a paper on bivalent dopamine agonists with cooperative binding and functional activity at D_2_R, with modulating effects on alpha-synuclein protein aggregation and toxicity. The structures studied were a hybrid of pramipexole and **P1**, linked by various linkers [69]. In preceding studies, an increase in potency was achieved with the optimal length of the methylene linker of 7–10 methylene units [72]. In this study, the linker was modified by inserting more rigid moieties and introducing functional groups on the aromatic moiety of the linker [69]. One of the phenyl moieties was replaced by a bioisosteric 2-aminothiazole moiety (**P4**) and affinity to both dopamine D_2_R and D_3_ receptor (D_3_R) was maintained (D_2_R IC_50_ = 13.4 nM, D_3_R IC_50_ = 13.3 nM). This compound appeared in seventh place in our screening, and in the described study [69], it turned out to be the most active structure. It was also one of the few compounds that aligned with all the pharmacophoric properties of the Phase pharmacophore model. The addition of hydroxyl groups at positions 1 and 4 to the aromatic linker ring in the presence of two isosteric 2-aminotiazole rings in the **P7** slightly reduced the affinity for D_2_R yet was still higher than for the reference molecule **P1** (IC_50_ = 34.37 nM vs. 41.0 nM). The hydroxyl groups themselves were well tolerated, while the presence of a thiazole-2-amino group in **P4** and **P7** had a moderate effect on the reduction of D_2_R potency. **P1** and **P7** met four out of five features of our pharmacophore model, both lacking one aromatic trait [69].

DPAT’s structure was also explored in another study [71], in which its 7-hydroxy derivative was modified by the addition of n-phenylpiperazine to the alkyl chain by a heterocyclic nitrogen atom, yielding **P6**, a potent D_2_R (EC_50_ = 9.98 nM) and D_3_R (EC_50_ = 2.91 nM) dual agonist. Further SAR studies discovered that the 2-(piperazin-1-yl)ethan-1-amine backbone is crucial for excellent dual activity. Bulky substituents (biphenyl or indole) at the piperazine N atom exhibit potent D_3_R activity, especially 2,3-dichlorophenyl. Propyl substitution at the alkyl amine increases the activity. 6-(5,6,7,8-tetrahydronaphthalen-1-ol)yl is more favorable for 6-(4,5,6,7-tetrahydrobenzo[d]thiazol-2-amine)yl, as well as an R to S configuration, for D_3_R binding. Overall, the most potent molecule (D_2_R EC_50_ = 0.87 nM, D_3_R EC_50_ = 0.23 nM) appeared to be (S)-6-((2-(4-(9H-carbazol-2-yl)piperazin-1-yl)ethyl)(propyl)amino)-5,6,7,8-tetrahydronaphthalen-1-ol, which is 2-(piperazin-1-yl)ethan-1-amine with 2-(9H-carbazol)yl substitution at the piperazine N atom and propyl and 6-(5,6,7,8-tetrahydronaphthalen-1-ol)yl substitution at the ethanamine atom [71].

One of the compounds (**P2**, propyl aminoindane), turned out to be a well-known compound that is an alkylated D_2_R agonist. It appeared in the study [70] as a molecule that, after appropriate modification (a biphenyl and an alline handle were attached to one of the N-propyl substituents of the aminoindane), served as a compound for the synthesis of two series of bidentate ligands selectively targeting D_2_R heterodimers.

**P3** and **P5** emerged from the study of the structure−functional−selectivity relationship of novel apomorphine analogues to develop selective D_1_R and D_2_R dual agonists, functionally biased toward activating the arrestin signaling pathway [66]. Overactivation of the G-protein pathway is associated with dyskinesias, while recruitment of β-arrestin 2 may not only desensitize the G-protein pathway but additionally activate the G-protein independent pathway, which can alleviate locomotor symptoms. Furthermore, both D_1_R and D_2_R activation are needed for a potent locomotor response. Compared to apomorphine, which is nonselective toward D_1_R (IC_50_ = 3.77 nM) and D_2_R (IC_50_ = 1.61 nM), propylnorapomorphine exhibits stronger affinity to D_1_R (IC_50_ = 1.1 nM) and D_2_R (IC_50_ = 0.04 nM), owing to elongation of the N-alkyl chain. O-acetylation of the catechol group of propylnorapomorphine yielded **P5**, which exhibits lesser affinity to D_1_R (IC_50_ = 31.7 nM) and maintains affinity to D_2_R (IC_50_ = 0.373 nM). Methylenedioxy protection of the catechol group of propylnorapomorphine yielded **P3**, which has an affinity to D_1_R (IC_50_ = 717.5 nM) and slightly lower to D_2_R (IC_50_ = 7.7 nM) compared to propylnorapomorphine. In the case of β-arrestin recruitment, apomorphine is biased toward recruitment for D_2_R (IC_50_ = 10.1 nM) rather than D_1_R (IC_50_ = 520.8 nM). In propylnorapomorphine, elongation of the N-alkyl chain further deepens this bias for D_2_R (IC_50_ = 1.18 nM) compared to D_1_R (IC_50_ = 1884 nM). O-acetylation of the catechol group in the case of **P5** slightly diminished recruitment for D_2_R (IC_50_ = 6.34 nM), while greatly lowering for D_1_R (IC_50_ = 5496 nM). Lastly, methylenedioxy protection of the catechol group of **P3** resulted in the inhibition of β-arrestin recruitment, both for D_2_R (IC_50_ = 520 nM) and for D_1_R (IC_50_ = 1949 nM) [66].

**P8**–**P10** [67] were described in a paper that investigated 2-phenylcyclopropylmethylamine (PCPMA) derivatives for partial agonism at the D_2_R. The **P8** compound was formed by the propylation of the secondary amino group in the PCPMA part. This did not result in an improvement in its activity (EC_50_ = 53.5 nM). **P10** with a chlorine atom on the phenyl ring in the para position relative to the methoxy substituent showed an increase in activity, had the best activity toward the D_2_R among all the new compounds in the entire study (EC_50_ = 2.63 nM), and, in our screening, had the highest activity [67].

Paying attention to the results, it seems that, for activity against D_2_R, the (S)-N6-(2,5-dimethyl-4-(2-(propylamino)ethyl)phenethyl)-N6-propyl-4,5,6,7-tetrahydrobenzo[d]thiazole-2,6-diamine backbone linked to a hydroxynaphthalene substituent via a second amine group might be a promising scaffold. Compounds that are modifications of propylapomorphine also showed good activity—beneficial here seems to be the O-acetylation of the catechol group, on the other hand. Moreover, 2-phenylcyclopropylmethylamine derivatives containing an (S)-N6-(2,5-dimethyl-4-(2-(propylamino)ethyl)phenethyl)-N6-propyl-4,5,6,7-tetrahydrobenzo[d]thiazole-2,6-diamine substituent also constitute a promising area for exploration. Here, the presence of a chlorine atom on the phenyl ring at the para position relative to the methoxy substituent proved most favorable for SAR. For dual D_2_R and D_3_R activity, structures based on an N-(2-(piperazin-1-yl)ethyl)propan-1-amine core, with bulky, hydrophobic substitutions at the heterocyclic N atom and 6-(5,6,7,8-tetrahydronaphthalen-1-ol)yl substitution at the alkyl N atom with R configuration, might find use as template structures.

#### 3.2.3. Serotonin Receptors

Serotonin receptors are divided into seven receptor families: 5-HT_1_, 5-HT_2_, 5-HT_3_, 5-HT_4_, 5-HT_5_, 5-HT_6,_ and 5-HT_7_. In total, at least 14 subtypes of these receptors have been discovered. All the families except 5-HT_3_Rs belong to the GPCRs. In turn, 5-HT_3_Rs are sodium–potassium ligand-gated ion channels [73]. 

The 5-HT_1_Rs and 5-HT_5_Rs are coupled to the G_i_/G_0_ protein; their activation causes a decrease in the intracellular cAMP concentration. 5-HT_2_Rs are coupled to the G_q11_ protein, and their activation causes an increase in the inositol trisphosphate (IP3) and diacylglycerol (DAG) concentrations. 5-HT_4_Rs, 5-HT_6_Rs, and 5-HT_7_Rs are coupled to the G_s_ protein; therefore, when activated, the cellular cAMP concentrations increase. All the receptor families are found in the CNS, where they are responsible for mood, learning, memory, sleep, locomotion, addiction, feelings of anxiety, or thermoregulation, among other things. Some are also found in the vascular system (5-HT_1_Rs, 5-HT_2_Rs, 5-HT_7_Rs) and gastrointestinal tract (5-HT_2_Rs, 5-HT_3_Rs, 5-HT_4_Rs), while 5-HT_2_Rs are also found in platelets and smooth muscle, and 5-HT_3_Rs are also found in the PNS [74].

##### 5-HT_1A_ Receptor

The 5-HT_1A_R is one of the best-studied serotonin receptors as the main serotonin inhibitory receptor in the brain. Two populations of this receptor can be distinguished—auto- and heteroreceptors. As an autoreceptor, it appears at presynaptic terminals in the sutural nuclei, where it controls the excitation of serotonergic neurons and the secretion of neurotransmitters. Heteroreceptors are expressed on non-serotonergic neurons, appearing mainly in the limbic system (body and dendrites of glutamatergic neurons, axons of γ-aminobutyric acid (GABA) neurons, or cholinergic neurons). Some receptors regulate the release of ACh (medial septum), glutamate (prefrontal cortex), or dopamine (midbrain cap) [73,75]. 

Because of its importance in the pathophysiology of neuropsychiatric disorders such as depression and anxiety [76], we chose it as a target for pharmacophore-based screening. 

Out of the structure of 5-HT_1A_R complexed with serotonin (agonist; PDB ID: 7E2Y), a pharmacophore model was proposed (Figure 8 and Figure 9, Table 14).

#### 3.2.4. 7E2Y (Serotonin 5-HT_1a_ Receptor) Pharmacophore Screening

From the ChEMBL database, we queried for molecules exhibiting an EC_50_ ≤ 100 nM of human 5-HT_1A_R activation in assays published in 2020 and later, then conducted pharmacophore-based ligand screening using the model proposed by the Phase module (Table 15).

The highest score value was calculated for **H1**, developed in the study exploring multitarget 5-HT_1A_R agonists and D_2_R, 5-HT_2A_ receptor (5-HT_2A_R) antagonists as schizophrenia drug candidates by automated deep-learning workflow. Typical antipsychotics are mainly D_2_R antagonists and exhibit good control of positive schizophrenia symptoms but cause various side effects like Parkinson-like extrapyramidal symptoms or tardive dyskinesia. Atypical antipsychotics usually exhibit inhibition (low affinity) toward D_2_R and (high affinity) 5-HT_2A_R, which facilitates less risk of side effects, but exhibit unsatisfactory control of cognitive dysfunction and negative symptoms. 5-HT_1A_R agonism may improve control of cognitive dysfunction and negative symptoms and alleviate side effects. The currently used atypical antipsychotics have a low ratio of 5HT_1A_R/D_2_R affinity and the higher ratio may improve the aforenoted pharmacodynamics of antipsychotics. The goal was to find novel structures, exhibiting high activity and low similarity, using deep-learning model assembly. To do so, two deep neural networks were built and then trained on data from the GLASS, Reaxys, and SciFinder databases. Out of the identified molecules, **H11** exhibited the strongest affinity toward 5-HT_1A_R. Its derivatization yielded **H1**, which exhibited the second-best 5-HT_1A_R affinity and emerged in pharmacophore screening (Table 16). It was noticed that similar compounds with a two-atom linker length had lower activities in relation to all three targets than compounds with a four-atom linker length. Fluorinated compounds displayed stronger agonism to 5-HT_1A_R [78]. 

Eight of the ten results of our screening (**H2**–**H9**) were described in one study [79], in which a new class of antipsychotic drugs was synthesized with a triazolopyridinone system linked to substituted piperazine or piperidine. The compounds obtained showed activity against 5-HT_1A_R (agonism), as well as 5-HT_2A_R and D_2_R (antagonism). The SAR for both serotonin and dopamine receptors was related to the variation of substituents on the triazolopyridinone ring and piperidine groups. **H3** with a triazolopyridinone scaffold showed good agonist activity on 5-HT_1A_R (EC_50_ = 1.7 nM) and high antagonistic activity against 5-HT_2A_R (IC_50_ = 34.2 nM) and against D_2_R (IC_50_ = 12.4 nM). To obtain further compounds, different substituents were introduced into the [1,2,4]triazolo [4,3-a]pyridin-3(2H)-one ring at positions 5–8. The introduction of halogen substituents into the ring at positions 6 and 8 resulted in a decrease in activity toward D_2_R and 5-HT_2A_R, while the activity toward the 5-HT_1A_R remained high, e.g., the **H4** captured by our screening expressed activity toward the 5-HT_1A_R of EC_50_ = 12.1 nM. Cyano and methoxy substituents were successively added to the above-mentioned ring at different positions as well. **H8** with the cyano substituent at position 5 was the least active against 5-HT_1A_R among the results of our screening (EC_50_ = 20.2 nM). On the other hand, **H6** with the cyano substituent at position 7 performed better in the biological tests (EC_50_ = 5.9 nM). **H5** with a 6-cyano substitution and **H7** with an 8-cyano substitution of [1,2,4]triazolo [4,3-a]pyridin-3(2H)-one achieved very good D_2_R inhibitory activity (IC_50_ = 1.03 nM and 1.5 nM, respectively) while maintaining, especially **H7**, good agonist activity on 5-HT_1A_R (EC_50_ = 9.7 nM and 1.4 nM). These results suggest the influence of the substituent position for this group of compounds. In **H9**, the thiophene ring was exchanged for a thiazole ring, and in **H2**, a fluorine substituent additionally appeared at position 8 of the thiazolpyrrolidine. This led to a sharp decrease in activity against 5-HT_2A_R and D_2_R for **H9** (IC_50_ = 117 nM and 2730 nM) while maintaining good activity against 5-HT_1A_R (EC_50_ = 2.8 nM). **H2** showed better activity at all three receptors and fantastic activity against the 5-HT_1A_R (EC_50_ = 0.1 nM).

The last hit described herein—**H10**—came from research focused on [50] the search for chimeric vilazodone-donepezil derivatives targeting 5-HT_1A_R, SERT, and acetylcholinesterase (AChE) [50]. Such compounds were expected to be an ideal response to depression co-occurring with Alzheimer’s disease. **H10** was the second most active against the 5-HT_1A_R (EC_50_ = 9.0 nM)—the introduction of vinyl instead of a methyl substituent at the 1-methylpiperidine moiety resulted in even higher potency (EC_50_ = 8.6 nM). Under the assumptions of the aforementioned work, compounds showing good activity on all three targets, including SERT and AChE, were found to be superior overall to **H10**, even though on the 5-HT_1A_R alone its activity was almost the best. Our model focused exclusively on 5-HT_1A_R, so it can be said that it did well in this screening by typing just this compound [50].

Summarizing the above studies, for activity toward the 5-HT_1A_R, as well as the dopamine D_2_R, compounds built on a 2-(4-(4-(4-(benzo[d]thiazol-4-yl)piperazin-1-yl)butyl)-[1,2,4]triazolo [4,3-a]pyridin-3(2H)-one backbone or one in which the thiazole is replaced by a thiophene ring are beneficial. The most favorable according to the local SAR analysis is the presence of a fluorine or cyano group at position 8 of the [1,2,4]triazolo [4,3-a]pyridin-3(2H)-one scaffold. The position of the substituent plays an important role in SAR in these compounds. Furthermore, a similar backbone to the above-described one, containing the stannous 2,2-dioxide 7-amino-1-methyl-3,4-dihydro-1H-benzo[c][1,2]thiazine instead of the 2-methyl-[1,2,4]triazolo [4,3-a]pyridin-3(2H) one, also shows good activity against 5-HT_1A_R and D_2_R. This dual-target interaction might also be deduced from the proposed pharmacophore models derived for both D_2_R and 5-HT_1A_R—in both cases, a positively ionizable feature, an aromatic feature as well as a hydrophobic or aromatic feature could be visible, and the ligands described herein align with these features (given the similar hydrophobic properties of aromatic rings and hydrophobic moieties). Moreover, it might also overlap with the SERT and MAO-B ones (Figure 10)—modifications of the compound **H10** provided herein provided activity not only for 5-HT_1A_R but also targets such as SERT and AChE.

### 3.3. Pharmacophore Model Alignment

Based on the publications described earlier, the search for multi-target ligands became an everyday practice. Of the targets we selected and for the studies described above, such combinations succeeded for the 5-HT_1A_R and D_2_R, and 5-HT_1A_R and SERT, pairs—where the designed molecules showed good activity more than once.

We decided to overlay the generated proposed models using Phase’s Hypothesis Alignment function to assess whether it would be possible to find molecules with multi-target activity on all, or at least most, of our chosen targets (Figure 10). Analyzing the pharmacophores, it can be seen that all four proposed pharmacophore models present common features: each contains a positively ionizable feature and at least one aromatic feature, and three of them also have a hydrophobic feature. When the models are overlaid, the positively ionizable features are in a fairly similar position, and the groups of aromatic and hydrophobic features also mostly overlap.

The similarity of the pharmacophore models derived from different target complexes suggests that it is fairly possible to obtain a multi-target structure, which could exhibit the desired activity on MAO-B, SERT, 5-HT_1A_R, and D_2_R, devoid of possible side effects, and potentially alleviate symptoms such as depression and motor dysfunctions. Based on the constructed models and their averaging, such a chemical structure would meet the characteristics of a pharmacophore for a D_2_R-based model—having one positive ionic feature, two aromatic features, and one hydrophobic feature, with a possibility of variation between the presence of additional hydrophobic or aromatic feature—with the distances between all the features roughly maintained (Figure 11).

## 4. Materials and Methods

A database search was conducted using the ChEMBL database [25]. The protein structures were obtained from the PDB, then prepared using the Maestro Schrödinger suite [80], using the Protein Preparation Wizard (default settings) [81,82]. The compounds’ structures were prepared using LigPrep (default settings) [83]. The pharmacophore models were generated using Phase (default settings unless otherwise specified). The pharmacophore screening was conducted using Phase (default settings) [22,23,84].

## 5. Conclusions

PeD has an extraordinarily strong impact on the lives of patients, and to the best of our knowledge, there are no specific medications and recommendations focused on this disease. Due to the similarity of the symptoms, treatment consists of administering the same drugs that are used in PD and providing only symptomatic treatment. Therefore, we see a lot of room for further research here, both for symptomatic drugs, which we have considered in this review, and for biological drugs that could change the fate of patients.

Therefore, herein, we reviewed section by section novel, active compounds that might either serve as hits for further optimization or candidates for further phases of studies, leading to potential use in the treatment of PeD. Due to the complex nature of the symptoms, we focused on several major therapeutic targets—the MAO-B, and SERT enzymes, as well as the D_2_R and the 5-HT_1A_R. We were able to propose pharmacophore models for each of the targets, which helped us to select, in our opinion, the compounds best suited in terms of the chemical structure, whose backbones are the newest directions from which medicinal chemistry can be further explored.

We believe that further research focused on multi-target ligands would be the most comprehensive approach for further symptomatic PeD treatment. Based on the analysis performed within the described studies and our proposed pharmacophore models, including their alignment, it is apparent that this is possible and can yield promising results. Especially, molecules based on the averaged/D_2_R model, which could exhibit an effect on all the studied targets—MAO-B, SERT, D_2_R, and 5-HT_1A_R—easing parkinsonism and depressive symptoms alike. In this context, the results of our comprehensive computer-aided analysis can be beneficial in finding such molecules, which proved safe and effective in preclinical and clinical studies, which would mean fewer drugs that a patient with PeD has to take at the same time, thus increasing the quality of the patient’s life.

Designing compounds that are multi-targeted carries numerous potential risks. Such compounds may act not only on the desired biological targets but also on off-targets. Subtypes of the same receptor family, in most cases, share a high level of structural similarity; therefore, the pharmacophores of the molecules acting on them might also overlap. This might drastically affect the selectivity of the desired compound over the main target. For instance, two of the serotonin receptors, namely 5-HT_2B_ and 5-HT_2C_, represent the best example of such. Due to the high dimensionality of the data, the complexity of the neurological pathways to tackle, numerous targets and off-targets to be considered, and similarities between pharmacophore models, the help of precise data analysis through artificial intelligence (AI) may prove useful in the further exploration of the increasingly expanding vast accessible chemical space [85,86].

While the significant challenges of finding drugs for rare diseases have already been described in Section 1.2, it is worth remembering that the search for any new drug to hit the market is a serious venture. Even if a substance will fit the proposed pharmacophore and would meet all of the hit/lead compound requirements, its journey from the laboratory bench to the patient’s bedside is exceptionally long. Once we have found (through in silico models) a molecule that should work on the screen, the path of such a potential drug might seem unending, starting from in vitro biological affinity and ADME-Tox studies, and extensive testing of the molecule’s physiochemical properties, with the next stop being in vivo models, primarily for toxicity. Only then, after being proven active and stable, do the three phases of clinical trials await, where the drug is evaluated for safety and efficacy first on healthy individuals, then on sick patients. At any stage of this long process, a drug candidate may not turn out to be good enough [87,88]. On top of this, clinical trials for rare genetic diseases face several additional challenges, such as difficulties in patient recruitment, gaps in basic research, ethical concerns, regulatory hurdles, or more down-to-earth matters such as economic profitability.

Thus arises the importance of translational research, especially in the search for orphan drugs. By assembling diverse multidisciplinary research teams, the process of translating basic research findings into novel therapies can be shortened. Additionally, bi-directional knowledge circulation between researchers, and clinical and social personnel has been suggested to speed up these breakthroughs [14].

Additionally, we can see that research on rare diseases and common diseases is interlocked more than was thought, in such a way that they can both fuel each other’s findings; e.g., research on better PD and depression medicines can improve PeD treatment and vice versa. Therefore, the search for the best possible therapy—in this case, for PeD—poses a particular challenge to complex research teams. In light of the use of increasingly sophisticated computational methods, a reality check, based on human knowledge, intuition, and communication, cannot be forgotten. Hence, the considerations described in our paper undoubtedly contribute to the development of these global and comprehensive efforts to improve PeD therapy.

## Figures and Tables

**Figure 1 ijms-25-10652-f001:**
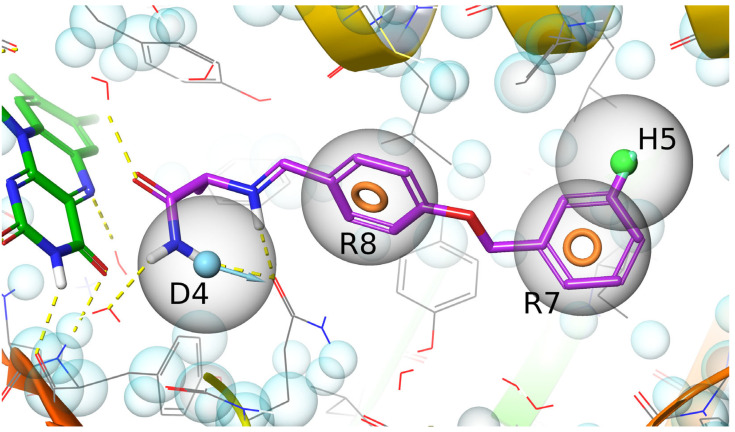
Safinamide (purple) complexed with MAO-B (gray) with, superimposed, the proposed pharmacophore model (balls and toruses). Dashed lines denote hydrogen bonds (yellow). Balls denote hydrogen bond donor feature (D, blue, arrow indicates bond direction) and hydrophobic feature (H, green), and toruses denote aromatic features (R, orange). Blue spheres denote excluded volumes. On the left, the flavin adenine dinucleotide (FAD) structure is visible (green).

**Figure 2 ijms-25-10652-f002:**
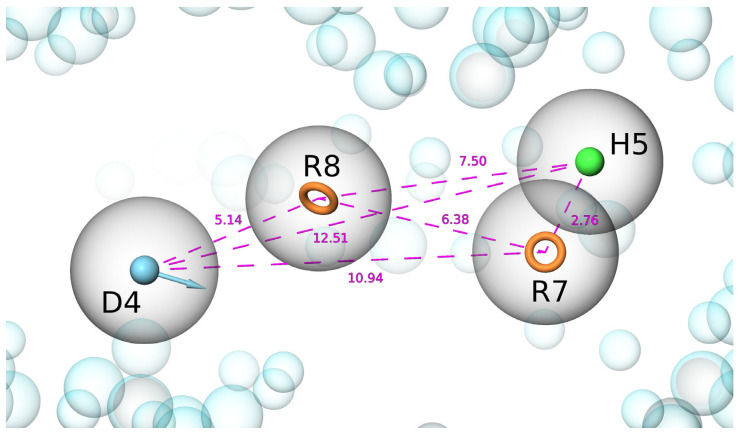
Proposed MAO-B pharmacophore model derived from the 2V5Z structure (balls and toruses). Balls denote hydrogen bond donor feature (D, blue, arrow indicates bond direction) and hydrophobic feature (H, green), and toruses denote aromatic features (R, orange). Dashed purple lines denote distances between features, with measurements in Å written beside them. Blue spheres denote excluded volumes.

**Figure 3 ijms-25-10652-f003:**
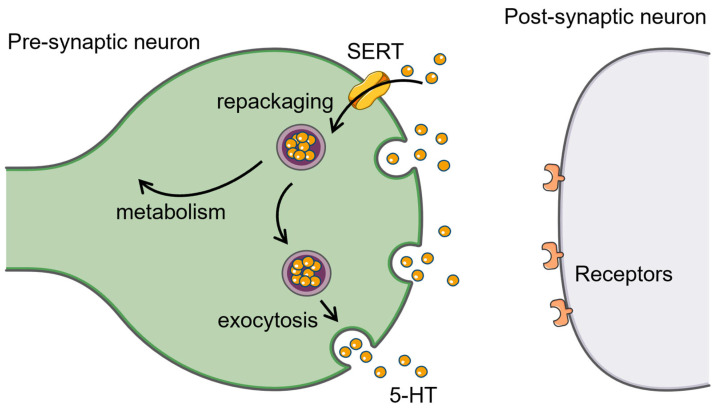
Model of a serotonergic synapse. SERT facilitates serotonin (5-HT, orange spheres) reuptake. Adapted from [42].

**Figure 4 ijms-25-10652-f004:**
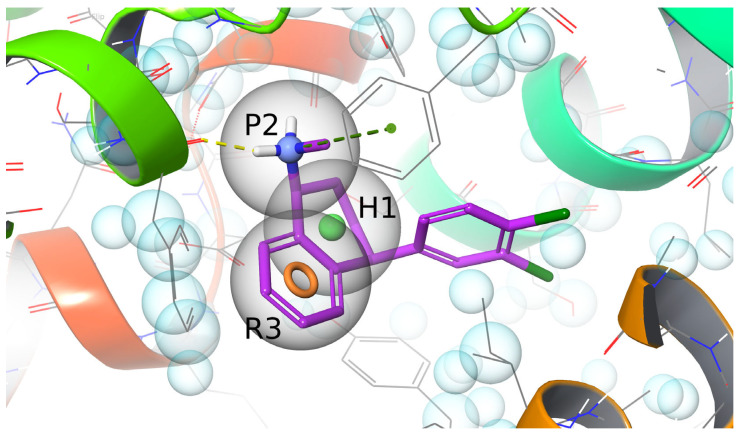
Sertraline (purple) complexed with SERT (gray) superimposed with the proposed pharmacophore model (balls and torus). Dashed lines denote hydrogen bond (yellow) and π-cation interaction (green). Balls denote positive ionic feature (P, blue), hydrophobic feature (H, green) and torus denotes aromatic feature (R, orange). Blue spheres denote excluded volumes.

**Figure 5 ijms-25-10652-f005:**
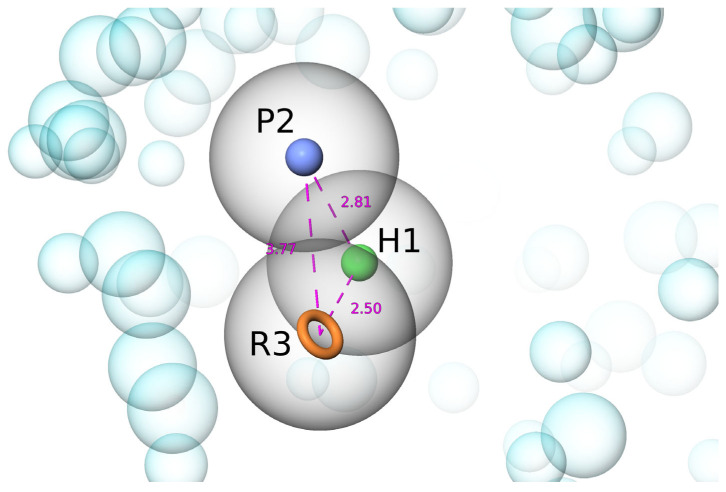
Proposed SERT pharmacophore model derived from the 6AWO structure (balls and torus). Balls denote positive ionic feature (P, blue) and hydrophobic feature (H, green), and torus denotes aromatic feature (R, orange). Dashed purple lines denote distances between features, with measurements in Å written beside them. Blue spheres denote excluded volumes.

**Figure 6 ijms-25-10652-f006:**
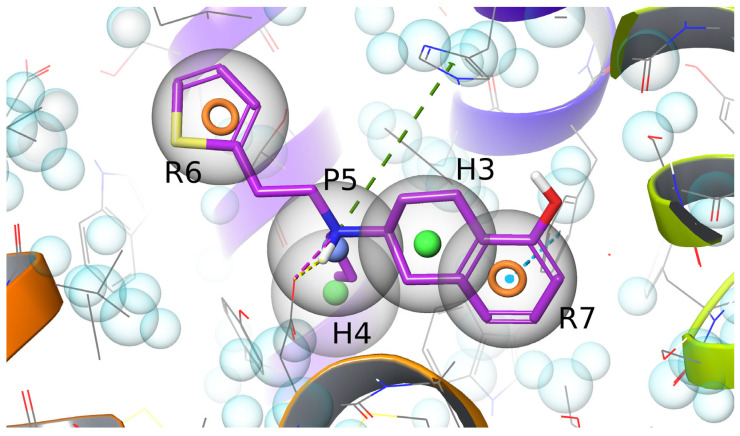
Rotigotine (purple) complexed with D_2_R (gray) with, superimposed, the proposed pharmacophore model (balls and toruses). Dashed lines denote hydrogen bond (yellow), salt bridge (red), π-π stacking (blue), and π-cation interaction (green). Balls denote positive ionic feature (P, blue) and hydrophobic features (H, green), and toruses denote aromatic features (R, orange). Blue spheres denote excluded volumes.

**Figure 7 ijms-25-10652-f007:**
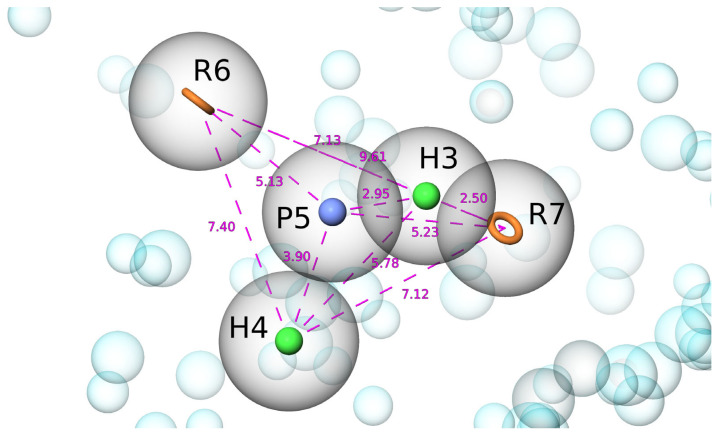
Proposed D_2_R pharmacophore model derived from the 8IRS structure (balls and toruses). Balls denote positive ionic feature (P, blue) and hydrophobic features (H, green), and toruses denote aromatic features (R, orange). Dashed purple lines denote distances between features, with measurements in Å written beside them. Blue spheres denote excluded volumes.

**Figure 8 ijms-25-10652-f008:**
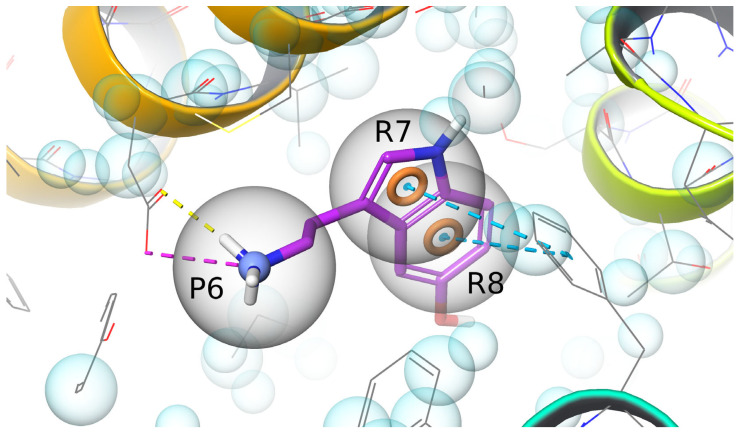
Serotonin (purple) complexed with 5-HT_1A_R (gray) with, superimposed, the proposed pharmacophore model (ball and toruses). Dashed lines denote hydrogen bond (yellow), salt bridge (red) and π-π stacking interactions (blue). Ball denotes positive ionic feature (P, blue) and toruses denote aromatic features (R, orange). Blue spheres denote excluded volumes.

**Figure 9 ijms-25-10652-f009:**
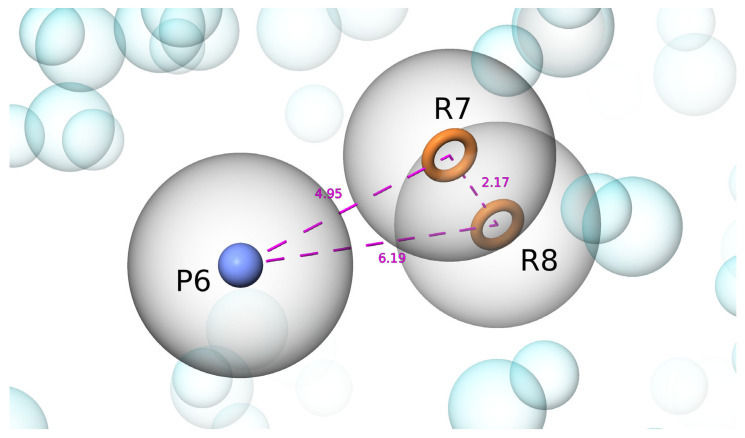
Proposed 5-HT_1A_R pharmacophore model derived from the 7E2Y structure (ball and toruses). Ball denotes positive ionic feature (P, blue) and toruses denote aromatic features (R, orange). Purple dashed lines denote distances between features, with measurements in Å written beside them. Blue spheres denote excluded volumes.

**Figure 10 ijms-25-10652-f010:**
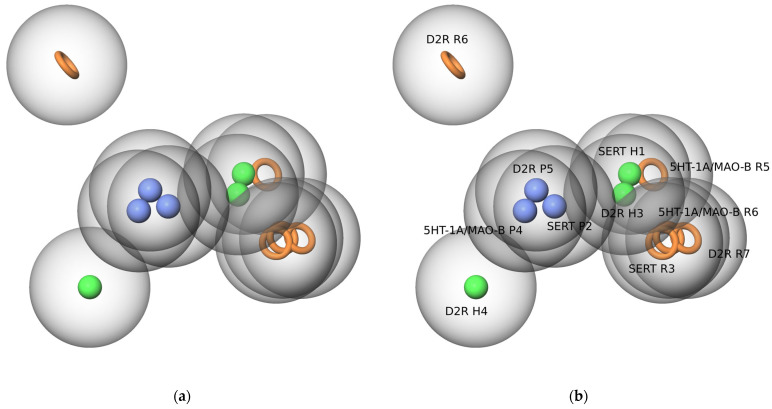
(**a**) Alignment of four proposed pharmacophore models (MAO-B, SERT, 5-HT_1A_R, D_2_R) used in this review (balls and toruses). (**b**) Balls denote positive ionic features (P, blue) and hydrophobic features (H, green), and toruses denote aromatic features (R, orange). The origin of each of the features has been described in capital letters.

**Figure 11 ijms-25-10652-f011:**
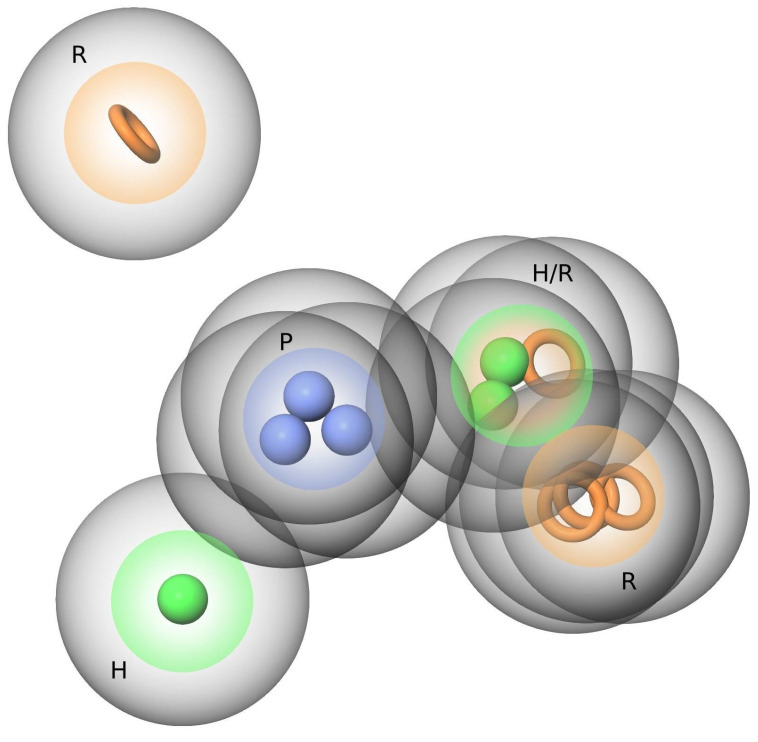
Averaged pharmacophore model based on the previously aligned models (balls and toruses). Balls denote positive ionic features (blue) and hydrophobic features (green), and toruses denote aromatic features (orange). Glowing spheres denote the averaged model’s features: positive ionic feature (P, blue), hydrophobic feature (H, green), hydrophobic or aromatic feature (H/R, green-orange gradient) and aromatic features (R, orange).

**Table 1 ijms-25-10652-t001:** Biological targets and their Protein Data Bank (PDB) entries used to develop the pharmacophore models.

Biological Target ^1^	PDB ID ^2^	Ligand ^3^	Ligand Type ^4^
MAO-B	2V5Z [27]	Safinamide	Antagonist
SERT	6AWO [28]	Sertraline	Antagonist
D_2_R	8IRS [29]	Rotigotine	Agonist
5-HT_1A_R	7E2Y [30]	Serotonin	Endogenous agonist

^1^ Targets include monoamine oxidase B (MAO-B), sodium-dependent serotonin transporter (SERT), dopamine D_2_ receptor (D_2_R), and serotonin 1A receptor (5-HT_1A_R). ^2^ PDB Identification Code (PDB ID) denotes a unique code under each molecular model deposited in the PDB. ^3^ Ligand denotes a small molecule bound to the target’s binding pocket. ^4^ Ligand type denotes the pharmacological type of the bound ligand.

**Table 2 ijms-25-10652-t002:** Overview of currently utilized MAO-B inhibitors.

Compound ^1^	ChEMBL ID ^2^	Structure	MAO-B IC_50_ [nM] ^3^
Selegiline	CHEMBL972	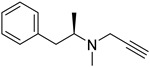	36.0 [35]
Rasagiline	CHEMBL887	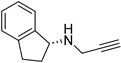	15.4 [36]
Safinamide	CHEMBL396778	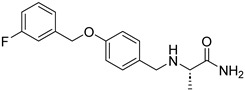	29.0 [37]

^1^ MAO-B inhibitors: selegiline, rasagiline, and safinamide, including ^2^ their unique ChEMBL database Identification Code (ChEMBL ID) and ^3^ half maximal inhibitory concentration (IC_50_), respectively.

**Table 3 ijms-25-10652-t003:** Distances (in Å) between the proposed MAO-B pharmacophore model features.

Pharmacophore Features	D4	H5	R7	R8
R8	5.14	7.50	6.38	
R7	10.94	2.76		
H5	12.51			
D4				

**Table 4 ijms-25-10652-t004:** MAO-B pharmacophore screening results, including up to 10 hit molecules, excluding well-established medicines and pharmacological tools.

Compound	ChEMBL ID	Structure	PhaseScreenScore	MAO-B IC_50_ [nM]
M1	CHEMBL4749026	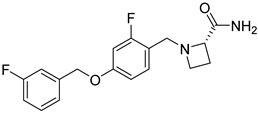	2.203314	26.0 [38]
M2	CHEMBL4792241	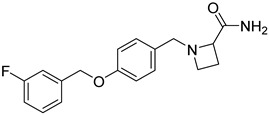	2.051776	46.0 [38]
M3	CHEMBL4747396	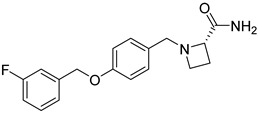	2.051776	21.0 [38]
Safinamide	CHEMBL396778	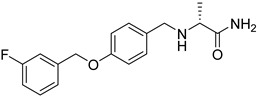	2.048958	25.0 [38]
M4	CHEMBL4750661	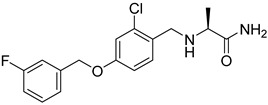	1.941374	28.0 [38]
M5	CHEMBL4763805	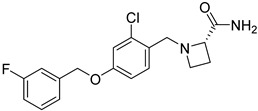	1.923999	35.0 [38]
M6	CHEMBL5077617	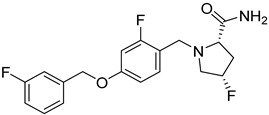	1.857744	30.0 [39]
M7	CHEMBL4752402	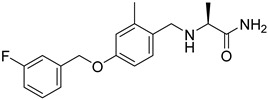	1.799135	69.0 [38]
M8	CHEMBL5083414	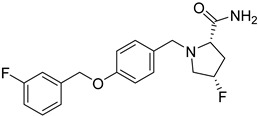	1.715590	19.0 [39]
M9	CHEMBL4743831	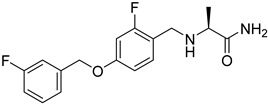	1.708420	61.0 [38]
M10	CHEMBL4860050	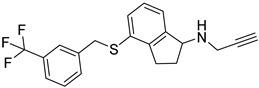	1.569462	4.7 [40]

**Table 5 ijms-25-10652-t005:** Well-known SERT inhibitors and their inhibitory potencies. The SERT inhibition values were obtained from [26].

Drug Class	Compound	Structure	SERT IC_50_ [nM]
SSRI ^1^	Paroxetine	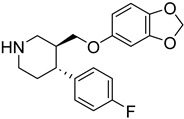	0.56
SSRI	Fluoxetine	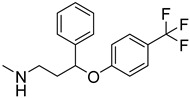	12.6
SSRI	Sertraline	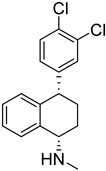	0.19
SSRI	Citalopram	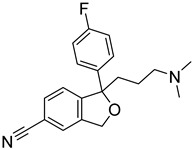	5.81
SSRI	Fluvoxamine	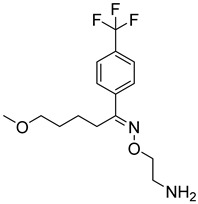	3.8
TCA ^2^	Clomipramine	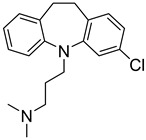	70.0
TCA	Imipramine	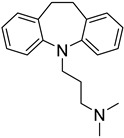	29.0
TCA	Amitriptyline	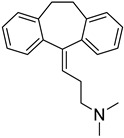	1.661
SNRI ^3^	Venlafaxine	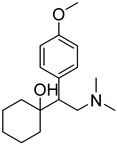	20.0
SARI ^4^	Trazodone	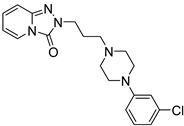	192.0

^1^ Selective serotonin reuptake inhibitor (SSRI). ^2^ Tricyclic antidepressant (TCA). ^3^ Serotonin–norepinephrine reuptake inhibitor (SNRI). ^4^ Serotonin antagonist and reuptake inhibitor (SARI).

**Table 6 ijms-25-10652-t006:** Distances (in Å) between the proposed SERT pharmacophore model features.

Pharmacophore Features	H1	P2	R3
R3	2.50	3.77	
P2	2.81		
H1			

**Table 7 ijms-25-10652-t007:** SERT pharmacophore screening results, including all the output molecules, since the screening results contain less than 10 hit molecules.

Compound	ChEMBL ID	Structure	PhaseScreenScore	SERT IC_50_ [nM]
Sertraline	CHEMBL809	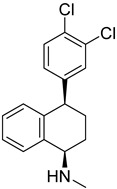	1.963006	0.19 [47]
Dextromethorphan	CHEMBL206132	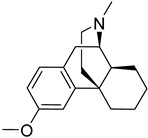	1.368854	56.0 [48]
Imipramine	CHEMBL11	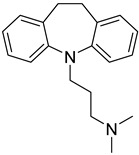	1.284248	29.0 [47]
S1	CHEMBL5175011	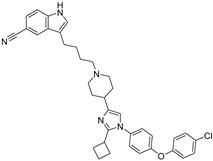	1.199585	7.47 [49]
S2	CHEMBL5086545	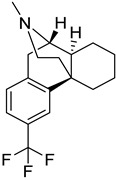	0.924859	60.0 [48]
S3	CHEMBL5081803	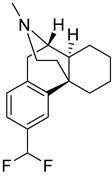	0.924859	24.0 [48]
S4	CHEMBL5093316	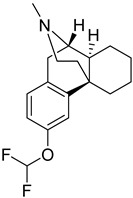	0.924859	55.0 [48]
S5	CHEMBL5078388	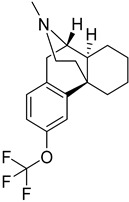	0.924859	31.0 [48]
Citalopram	CHEMBL549	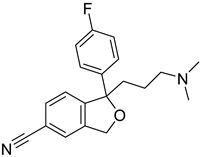	0.751749	5.81 [50]
S6	CHEMBL5207764	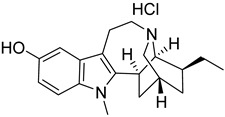	0.584015	59.0 [51]
S7	CHEMBL5188930	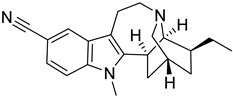	0.584015	5.1 [51]
S8	CHEMBL5201219	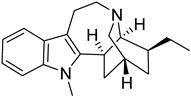	0.584015	80.0 [51]
S9	CHEMBL5175119	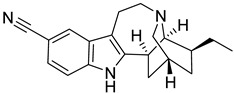	0.583134	26.0 [51]
Lumateperone	CHEMBL3306803	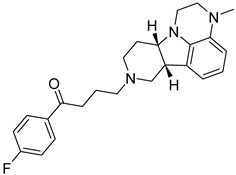	0.490147	3.3 [52]

**Table 8 ijms-25-10652-t008:** Notable compounds explored in the dual receptor for advanced glycation end products (RAGE)/SERT inhibitors study [49].

Compound	ChEMBL ID	Structure	RAGE IC_50_ [nM]	SERT IC_50_ [nM]
Azeliragon	CHEMBL3989929	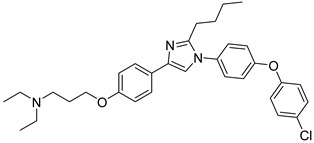	13,470	>3000
Vilazodone	CHEMBL439849	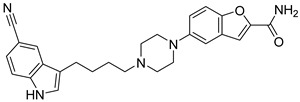	>200,000	0.40
S1	CHEMBL5175011	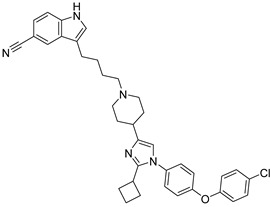	8290.0	7.47
S10	CHEMBL5188606	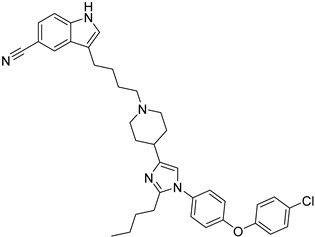	12,920.0	65.58
S11	CHEMBL5192104	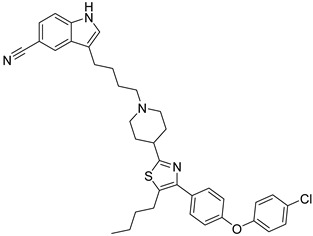	14,270	67.83
S12	CHEMBL5203206	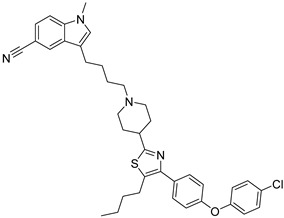	8260	31.09
S13	CHEMBL5191418	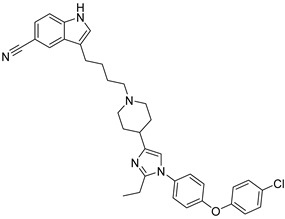	3490	4.40
S14	CHEMBL5208902	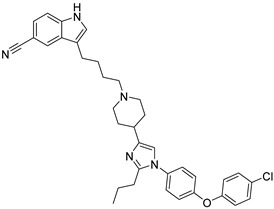	12,490	7.77
S15	CHEMBL5175011	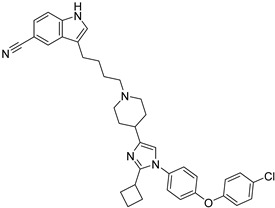	8290	7.47
S16	CHEMBL5194592	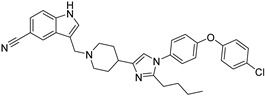	4040	57.73
S17	CHEMBL5180815	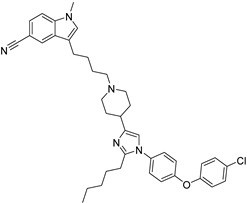	6030	15.05

**Table 9 ijms-25-10652-t009:** Notable compounds explored in the fluoroalkylation of dextromethorphan study [48].

Compound	ChEMBL ID	Structure	σ1 *K*_i_ ^1^ [nM]	σ2 *K*_i_ [nM]	NMDA *K*_i_ [nM]	SERT IC_50_ [nM]
Dextromethorphan	CHEMBL206132	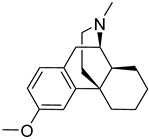	73	862	624	56.0
AVP-786	CHEMBL5078675	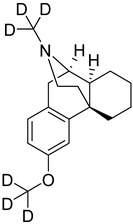	n.t. ^2^	n.t.	n.t.	n.t.
S2	CHEMBL5086545	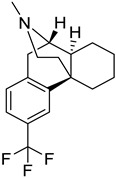	813	885	>10,000	60.0
S3	CHEMBL5081803	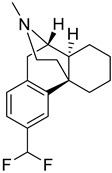	145	353	>10,000	24.0
S4	CHEMBL5093316	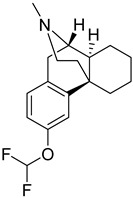	568	1281	>10,000	55.0
S5	CHEMBL5078388	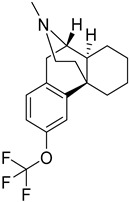	757	833	>10,000	31.0

^1^ Inhibition constant (*K*_i_). ^2^ Not tested (n.t.).

**Table 10 ijms-25-10652-t010:** Notable compounds described in the patent for ibogaine and its analogues [51].

Compound	ChEMBL ID	Structure	VMAT2 IC_50_ [nM]	SERT IC_50_ [nM]
Ibogaine	CHEMBL1215855	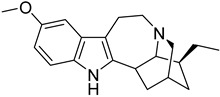	4000.0	500.0 [51]
Noribogaine	CHEMBL5202868	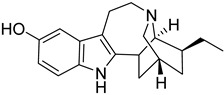	570.0	280.0 [51]
S6	CHEMBL5207764	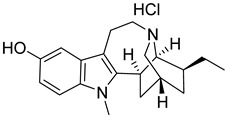	170.0	59.0 [51]
S7	CHEMBL5188930	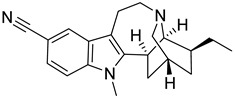	440.0	5.1 [51]
S8	CHEMBL5201219	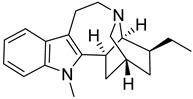	1500.0	80.0 [51]
S9	CHEMBL5175119	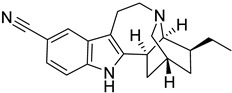	3300.0	26.0 [51]

**Table 11 ijms-25-10652-t011:** D_2_-like receptor agonists indexed in ChEMBL.

Compound	Structure	D_2_R EC_50_ ^1^ [nM]
Rotigotine	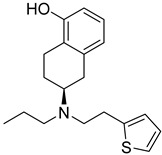	121.6
Apomorphine	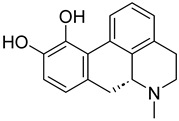	1542.7
Pramipexole	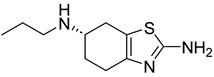	14000.1
Ropinirole	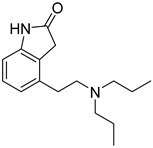	7999.4
Bromocriptine	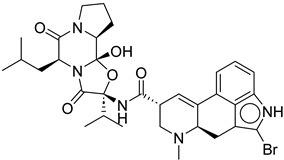	27.8

^1^ Half maximal effective concentration (EC_50_).

**Table 12 ijms-25-10652-t012:** Distances (in Å) between the proposed D_2_R pharmacophore model features.

Pharmacophore Features	H3	H4	P5	R6	R7
R7	2.50	7.12	5.23	9.61	
R6	7.13	7.40	5.13		
P5	2.95	3.90			
H4	5.78				
H3					

**Table 13 ijms-25-10652-t013:** D_2_R pharmacophore screening results, including up to 10 hit molecules, excluding well-established medicines and pharmacological tools.

Compound	ChEMBL ID	Structure	PhaseScreenScore	D_2_R EC_50_ [nM]
Propylnor-apomorphine	CHEMBL225230	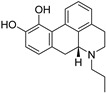	2.148715	1.175 [66]
Quinpirole	CHEMBL240773	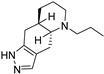	2.135124	1.18 [67]
Pramipexole	CHEMBL301265	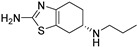	2.102299	6.457 [68]
P1 (5-OH-DPAT)	CHEMBL273273	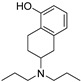	2.080779	41.0 [69]
P2	CHEMBL4781480	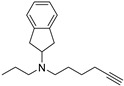	2.032088	3.4 [70]
P3	CHEMBL4470553	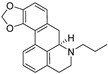	2.016313	7.7 [66]
P4	CHEMBL5267221	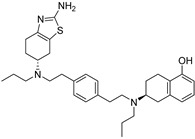	1.990749	13.4 [69]
P5	CHEMBL4555547	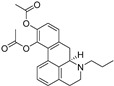	1.916507	0.373 [66]
P6	CHEMBL458088	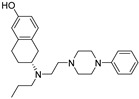	1.819042	9.98 [71]
P7	CHEMBL5266134	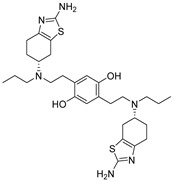	1.801997	34.37 [69]
P8	CHEMBL4846101	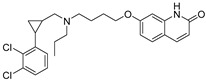	1.798176	53.5 [67]
P9	CHEMBL4875081	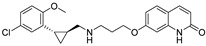	1.765344	3.41 [67]
P10	CHEMBL4846472	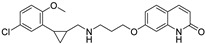	1.765344	2.63 [67]

**Table 14 ijms-25-10652-t014:** Distances (in Å) between the proposed 5-HT_1A_R pharmacophore model features.

Pharmacophore Features	P6	R7	R8
R8	6.19	2.17	
R7	4.95		
P6			

**Table 15 ijms-25-10652-t015:** 5-HT_1A_R pharmacophore screening results, including up to 10 hit molecules, excluding well-established medicines and pharmacological tools.

Compound	ChEMBL ID	Structure	PhaseScreenScore	5-HT_1A_ EC_50_ [nM]
Serotonin	CHEMBL39	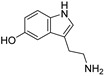	2.890	10.0 [77]
H1	CHEMBL4751542	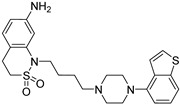	2.103	0.7943 [78]
H2	CHEMBL4633397	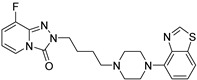	2.030	0.1 [79]
H3	CHEMBL4638599	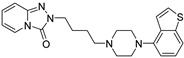	2.020	1.7 [79]
H4	CHEMBL4636321	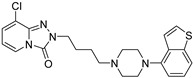	2.010	12.1 [79]
H5	CHEMBL4644391	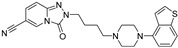	1.998	9.7 [79]
H6	CHEMBL4648979	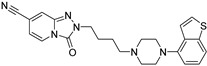	1.998	5.9 [79]
H7	CHEMBL4638430	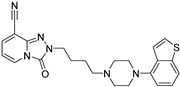	1.997	1.4 [79]
H8	CHEMBL4644742	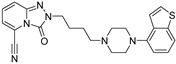	1.997	20.2 [79]
H9	CHEMBL4635890	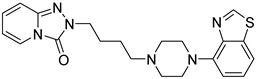	1.874	2.8 [79]
H10	CHEMBL5079986	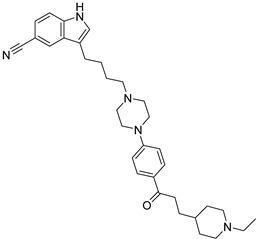	1.790	9.0 [50]

**Table 16 ijms-25-10652-t016:** Two most potent 5-HT_1A_R agonists from a multitarget schizophrenia drug study [78].

Compound	ChEMBL ID	Structure	D_2_ IC_50_ [nM]	5-HT_2A_ IC_50_ [nM]	5-HT_1A_ EC_50_ [nM]
H1	CHEMBL4751542	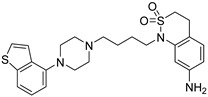	67.61	2.818	0.7943
H11	CHEMBL4741908	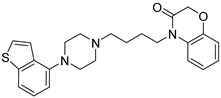	216.0	1.64	0.51

## Data Availability

The original contributions presented in this study are included in the article and Appendix A; further inquiries can be directed to the corresponding author.

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
