# Peer review of "Perry Disease: Current Outlook and Advances in Drug Discovery Approach to Symptomatic Treatment"

_ijms, 2024, doi:10.3390/ijms251910652_

Round 1
Reviewer 1 Report
Comments and Suggestions for Authors
This study titled by "Perry Disease: Current Outlook and Advances in Drug Discovery" provides a valuable and comprehensive review of PeD and potential therapeutic targets. However, a major revision are necessary to address redundancy, enhance data presentation, expand the discussion on clinical translation. Here are my concerns:
1, The descriptions of neurodegenerative diseases (NDDs) in the introduction and the section on rare diseases overlap significantly.
2, While the manuscript discusses potential therapeutic compounds, it lacks a detailed discussion on the clinical translation of these findings. More information on the challenges and strategies for moving from bench to bedside would enhance the manuscript. For example, the process of moving from in silico models to clinical trials is not addressed.
3, Table 1 and Table 2 could be better formatted to enhance readability and ensure consistency. The description of biological targets and their PDB entries could be clearer. Ensuring consistent formatting and clear separation of data would improve readability.
4, The manuscript primarily synthesizes existing knowledge without introducing significant new insights or hypotheses. A more critical analysis and discussion of potential future research directions would add value.
Author Response
Dear Reviewers,
First of all, we would like to thank you for peer revision and valuable comments on our manuscript. We have sincerely considered all of the received feedback and modified the text in response to the extensive and insightful reviewer suggestions. We hope that these changes will comply with the referees’ remarks and that the revised version of the manuscript is now suitable for the International Journal of Molecular Sciences.
All modifications we performed within the descriptions and graphics are presented in the revised manuscript in “tracking changes” form.
Enclosed pleased find our responses to Reviewers’ comments point by point as follows:
Review 1
- The descriptions of neurodegenerative diseases (NDDs) in the introduction and the section on rare diseases overlap significantly.
The introduction part of the manuscript has been modified so as to remove the overlapping parts of the text.
- While the manuscript discusses potential therapeutic compounds, it lacks a detailed discussion on the clinical translation of these findings. More information on the challenges and strategies for moving from bench to bedside would enhance the manuscript. For example, the process of moving from in silico models to clinical trials is not addressed.
The manuscript has been updated with information about the challenges and possible strategies that might be used in complimentary drug design. Yet, this work aimed to review the latest data specifically from the medicinal chemistry and computer-aided drug design (CADD) points of view. Therefore, the next stages of drug discovery and development, such as clinical trials have been omitted. However, due to the high importance of being able to translate laboratory tests into clinical trials, especially in the case of orphan diseases, we have expanded the manuscript with relevant information and updated the list of cited literature with corresponding papers.
- Table 1 and Table 2 could be better formatted to enhance readability and ensure consistency. The description of biological targets and their PDB entries could be clearer. Ensuring consistent formatting and clear separation of data would improve readability.
The tables were re-formatted to fit Reviewer’s suggestions. Other table captions were thoroughly checked as well.
- The manuscript primarily synthesizes existing knowledge without introducing significant new insights or hypotheses. A more critical analysis and discussion of potential future research directions would add value.
The manuscript has been updated with potential information on possible new structures, future research goals, chemical structure directions, and possible obstacles in finding new therapies for orphan diseases.
Reviewer 2 Report
Comments and Suggestions for Authors
This review summaries the latest updates about the Perry disease treatments focusing on medicinal chemistry and computer-aided drug design (CADD). Authors have found some proteins that can be the potential targets for the symptomatic treatment of the disease, such as monoamine oxidase B (MAO-B), serotonin transporter (SERT), dopamine D2 (D2R), and serotonin 5-HT1A (5-HT1AR) receptors. Further, authors reported some of the candidate compounds using Phase pharmacophore modeling software implemented in Schrödinger Maestro. I enjoyed the reading and found no mistake. Review is very well written and I recommend acceptance in its original form.
Author Response
Dear Reviewers,
First of all, we would like to thank you for peer revision and valuable comments on our manuscript. We have sincerely considered all of the received feedback and modified the text in response to the extensive and insightful reviewer suggestions. We hope that these changes will comply with the referees’ remarks and that the revised version of the manuscript is now suitable for the International Journal of Molecular Sciences.
All modifications we performed within the descriptions and graphics are presented in the revised manuscript in “tracking changes” form.
Enclosed pleased find our responses to Reviewers’comments point by point as follows:
Review 2
This review summaries the latest updates about the Perry disease treatments focusing on medicinal chemistry and computer-aided drug design (CADD). Authors have found some proteins that can be the potential targets for the symptomatic treatment of the disease, such as monoamine oxidase B (MAO-B), serotonin transporter (SERT), dopamine D2 (D2R), and serotonin 5-HT1A (5-HT1AR) receptors. Further, authors reported some of the candidate compounds using Phase pharmacophore modeling software implemented in Schrödinger Maestro. I enjoyed the reading and found no mistake. Review is very well written and I recommend acceptance in its original form.
We would like to sincerely thank the Reviewer 2 for time devoted to read our manuscript and all positive words about both, content and form of our review article.
Round 2
Reviewer 1 Report
Comments and Suggestions for Authors
Accept